# Prediction of Minimum Night Flow for Enhancing Leakage Detection Capabilities in Water Distribution Networks

Sang Soo Lee [1], Ho-Hyun Lee [2] and Yun-Jung Lee [1,*]

1    School of Electronic and Electrical Engineering, Kyungpook National University, Daegu 41566, Korea; sangsoo@kwater.or.kr
2    K-Water Research Institute, Korea Water Resources Corporation, Daejeon 34045, Korea; lhh@kwater.or.kr
*    Correspondence: yjlee@ee.knu.ac.kr; Tel.: +82-53-950-6562; Fax: +82-53-940-8862

**Abstract:** In South Korea, a water supply enhancement project is being carried out to preemptively respond to drought and water loss by reducing pipeline leakages and supplying stable tap water through the maintenance of an aging water supply network. In order to reduce water leakage, a District Metered Area (DMA) was established to monitor and predict the minimum night flow based on flow data collected from IoT sensors. In this study, a model based on Multi-Layer Perceptron (MLP) and Long Short-Term Memory (LSTM) was constructed to predict the MNF (minimum night flow) of County Y. The prediction of MNF results was compared with the MLP networks and the LSTM model. The outcome showed that the LSTM-MNF model proposed in this study performed better than the MLP-MNF model. Therefore, the research methods of this study can contribute to technical support for leakage reductions by preemptively responding to the expected increase in leakage through the prediction of the minimum flow at night.

**Keywords:** LSTM; water leakage; minimum night flow; waterworks; water network

## 1. Introduction

According to statistics provided by the Ministry of Environment in 2019, the total amount of water leakage in South Korea equates to 1 million $m^3$. The revenue water ratio is the ratio of the amount of water received as a fee out of the total amount of water produced at the water purification plant. Non-revenue water (NRW) represents the difference between water supplied and water sold, expressed as a percentage of net water supplied [1]. Countries such as the United States (12.78%) in 2011 and Poland (15.45%) in 2010 indicated low NRW, while Chile (33.3%) and Turkey (58.93%) in 2008 had high NRW [2]. In order to lower the NRW in South Korea, which has reached about 10.5% nationwide [3], actions such as detecting leaks, monitoring and analyzing flowmeters in District Metered Areas (DMAs), performing district meter analysis, managing distribution system pressure, preventing illicit water consumption, and training and education in waterworks management are being taken.

A DMA is defined as a discrete sector of a distribution network which is formed naturally or imposed, and can effectively evaluate the continuous flow of water supply through a flow meter installed at metering points, as shown in Figure 1.

The MNF in this study refers to the flow into the DMA in the middle of the night when water demand is at its lowest. The MNF is a common method used to evaluate water loss in a water network. The MNF includes the water demand at night and water leakage, as shown in Figure 2. Generally, very little water is used during night hours. The water demand at night means very little water is used, e.g., the water demand of toilet flushing late at night. When the amount of water demand is relatively small, the amount of water leakage can be predicted and analyzed by measuring the flow when the amount of water flowing into the area is minimized.

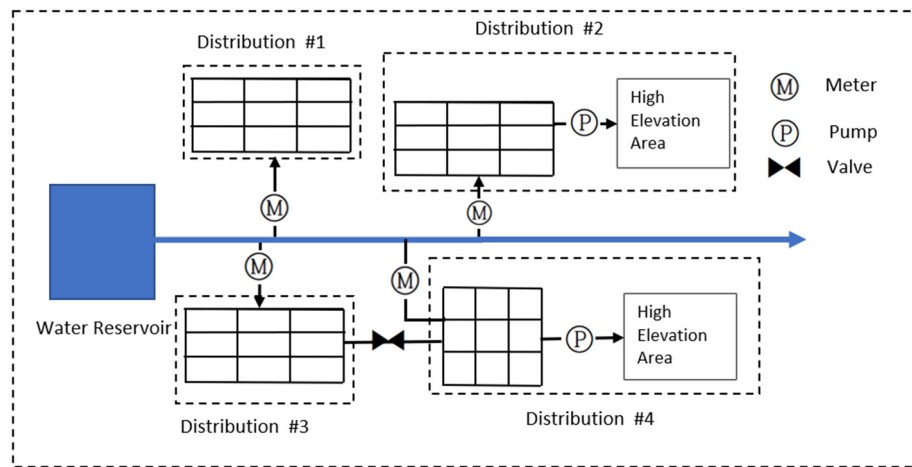

**Figure 1.** DMA (District Metered Area) Schematics.

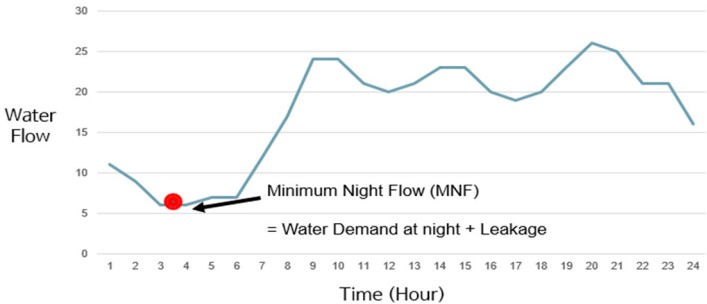

**Figure 2.** Configuration of the minimum night flow.

As the leakage continues, the losses such as supply and pressure also increase. Recently, Neuro Fuzzy has been used to predict leaks in water pipe networks [4–6] and Long Short-Term Memory (LSTM) has been used to predict water flood, water quality, and consumption [7–10]. In this paper, using the flow data acquired from the pipe network monitoring system based on Information and Communication Technology (ICT), the predicted values calculated by LSTM and Multi-Layer Perceptron (MLP) are compared with the actual minimum night flow.

## 2. Introduction and Application of Modeling

### 2.1. Sectorized Area Overview

The flow rates of three sectorized DMAs (YD 2, YD 3, YD 4) in the water distribution network of County Y were used in the modeling, as shown in Figure 3. There were very few customers using water at night in YD 2, YD 3, and YD 4. Single-jet water meters and digital water meters were installed in each household in the DMAs with advanced metering infrastructure (AMI). The water data, such as the flow and pressure, were acquired from supervisory control and data acquisition (SCADA) systems with tele metric and tele control (TM/TC) using wireless communication.

As shown in Figure 4, the flow of each DMA was measured by installing the LF-200 model electronic flowmeter made by Woori Technology (Korea), which has high accuracy considering the straightness and length of the pipe. The problem of low accuracy in MNF was solved by reducing the diameter of the water supply pipe in each DMA (YD 2: D200→D150; YD 3: D200→D150; YD 4: D250→D150) and increasing the speed of the water.

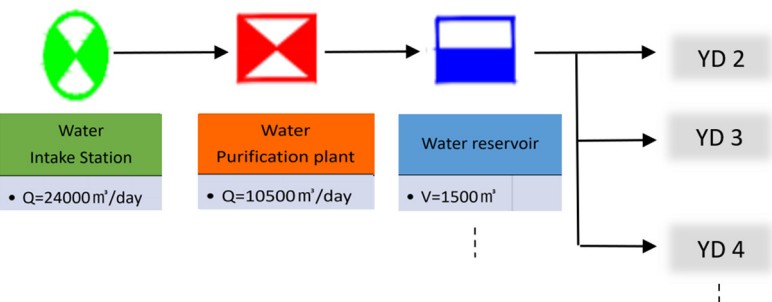

**Figure 3.** Schematic diagram of the water distribution network in County Y. The names of the sites are shown in Table 1.

**Table 1.** Information on the DMAs in YD 2, YD 3, and YD 4 (2011).

| DMA | Number of Households with Drinking Water Taps | Water Demand ($m^3$/day) | Pipe Length (km) |
|---|---|---|---|
| YD 2 (D150) | 1115 | 621 | 10.48 |
| YD 3 (D150) | 783 | 490 | 7.21 |
| YD 4 (D150) | 845 | 517 | 4.77 |

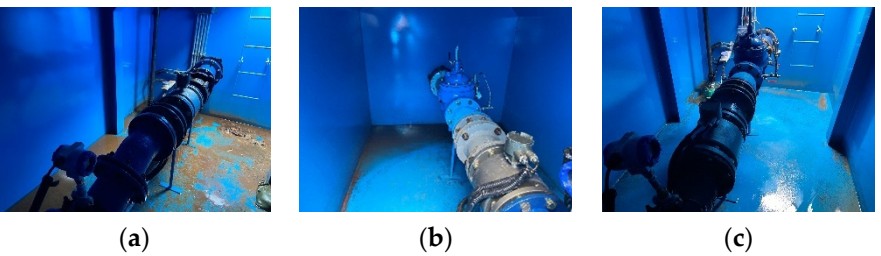

(**a**)  (**b**)  (**c**)

**Figure 4.** The electromagnetic flow meters in County Y: (**a**) YD 2, (**b**) YD 3, and (**c**) YD 4.

### 2.2. Framework of the Proposed Model

The flow data used in this study were obtained from 07/01/21 to 12/15/21 considering the learning time for training. Missing data from 12/16/21 to12/31/21 were excluded for precise prediction.

There were several minor instances of missing data, which were replaced with the data from the same time on the previous day. However, this did not significantly affect the training results. Additionally, flow data during any national holidays were included for training. Flow data acquired from the tele metric and tele control systems were consistent because they were acquired using serial communication. During this period, 70% of the data were used as the training set and the remaining 30% were used as the test set. A predictive model for 42 days (11/04/21~12/15/21) was used as the test set. The observation periods were related to the same dates.

For the flow prediction model applied in this paper, as shown in Table 2 and Figure 5, hourly flow data for the previous 7 days were used as input variables with LSTM/MLP and the hourly flow data of the next day were used output variables.

The method for extracting the MNF was finding the smallest flow data between 0 am and 5 am in terms of the hourly flow from the previous predictions, and the smallest flow measured in a 1-h interval was used as the minimum night flow [11–13].

**Table 2.** Design specifications with LSTM/MLP modeling variables.

| Item | Design Specifications |
|---|---|
| Input variables | Hourly flow for the previous 7 days |
| Output variables | Hourly flow next day |

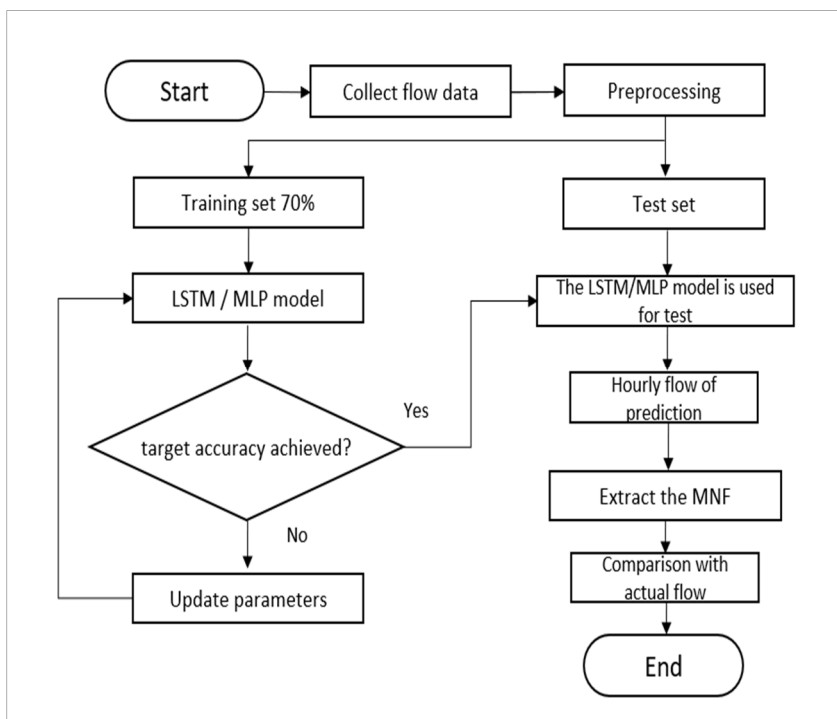

**Figure 5.** Flow chart of MNF prediction model.

### 2.3. MLP and LSTM Network Model

　　The MLP and LSTM networks were used to predict the hourly flow. The MLP network has an input layer, output layer, and several hidden layers, and is a kind of multi-layer feed-forward network based on the backpropagation algorithm during training as shown in Figure 6a. As shown in Figure 6b, LSTM networks are a type of recurrent neural network capable of learning order dependence in sequence prediction problems. A common LSTM unit is composed of a cell, an input gate, an output gate, and a forget gate. The parameters of the LSTM model were 168 units and 300 epochs. In addition, the activation function was considered as the rectified linear unit.

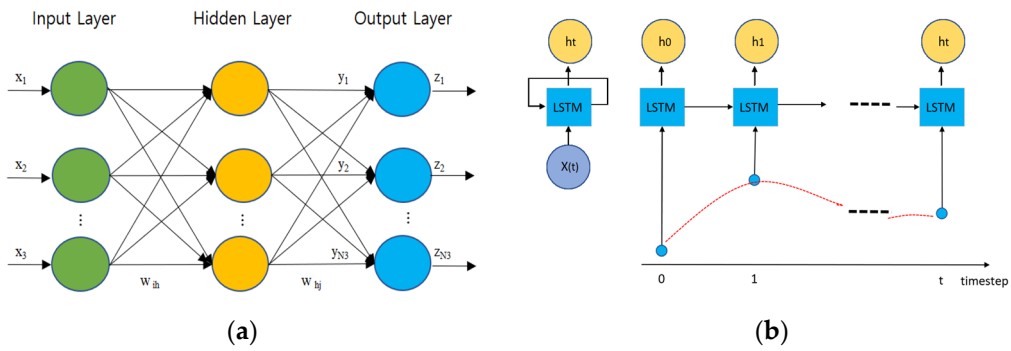

| (a) | (b) |

**Figure 6.** Schematic structure: (**a**) MLP neural network and (**b**) LSTM network.

　　Mean Absolute Percentage Error (MAPE), Mean Absolute Error (MAE), Root Mean Square Error (RMSE), and R-squared were considered for the performance evaluation:

$$\text{MAPE} = \frac{1}{N} \sum_{i=1}^{N} \left| \frac{Observed\ MNF_i - Predicted\ MNF_i}{Observed\ MNF_i} \right| \times 100 \tag{1}$$

$$\text{MAE} = \frac{1}{N} \sum_{i=1}^{N} |Observed\ MNF_i - Predicted\ MNF_i| \tag{2}$$

$$RMSE = \sqrt{\frac{1}{N} \sum_{i=1}^{N} (Observed\ MNF_i - Predicted\ MNF_i)^2} \tag{3}$$

## 3. Results

### 3.1. Case of YD 2

In this study we used KERAS, which is an open-source software library that provides a python interface for neural networks. Figures 7 and 8 present the MNF predictions with the test data from YD 2 with the MLP and LSTM models. As shown in Figures 7 and 8, the coefficient of determination was 0.868 for the MLP-MNF model and 0.907 for the LSTM-MNF model.

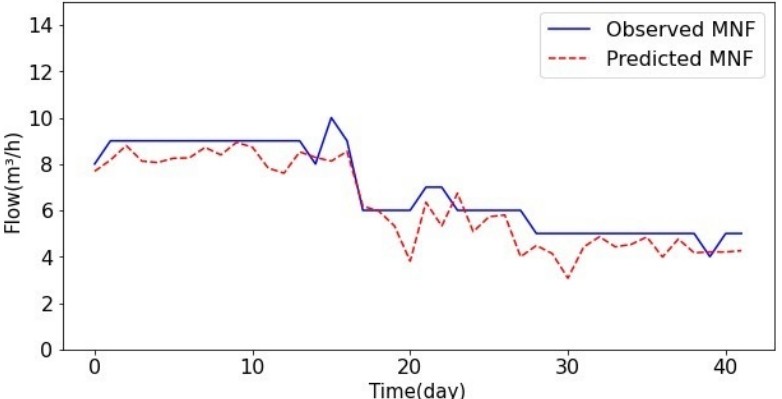

**Figure 7.** MNF prediction with the test data from YD 2 with the MLP model.

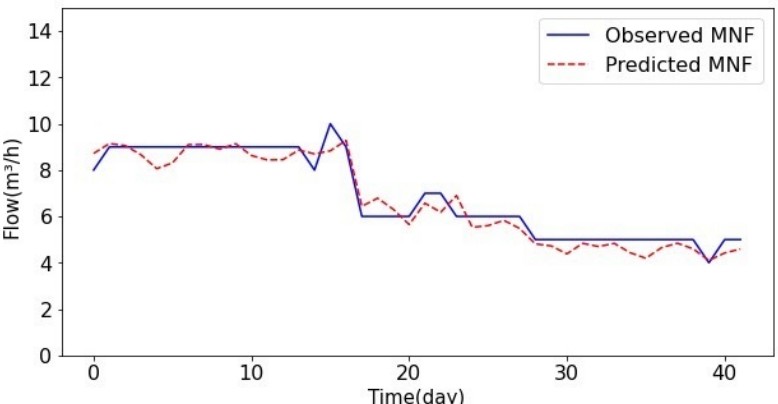

**Figure 8.** MNF prediction with the test data from YD 2 with the LSTM model.

### 3.2. Case of YD 3

Figures 9 and 10 show the MNF predictions with the test data from YD 3 with the MLP and LSTM models. As shown in Figures 9 and 10, the coefficient of determination was 0.257 for the MLP-MNF model and 0.488 for the LSTM-MNF model. The LSTM-MNF model clearly demonstrates where the flow changed rapidly.

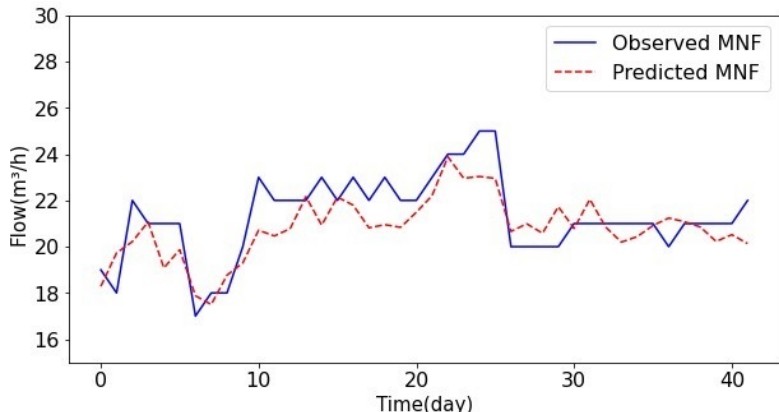

**Figure 9.** MNF prediction with the test data from YD 3 with the MLP model.

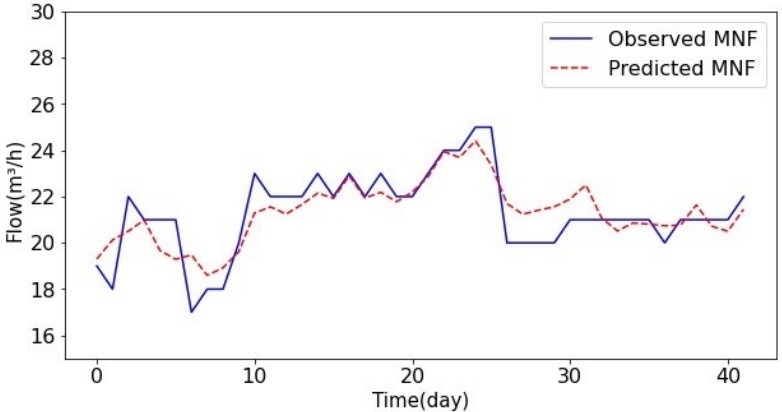

**Figure 10.** MNF prediction with the test data from YD 3 with the LSTM model.

### 3.3. Case of YD 4

Figures 11 and 12 show the MNF predictions with the test data from YD 4 with the MLP and LSTM models. As shown in Figures 11 and 12, the coefficient of determination was 0.637 for the MLP-MNF model and 0.733 for the LSTM-MNF model. There does not seem to be any significant difference between the two models; however, the MAPE of the LSTM-MNF model was 3.806 compared to 4.253 in the MLP-MNF model.

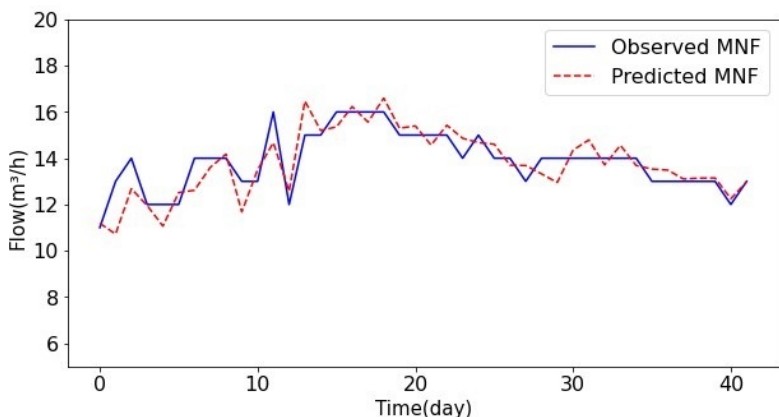

**Figure 11.** MNF prediction with the test data from YD 4 with the MLP model.

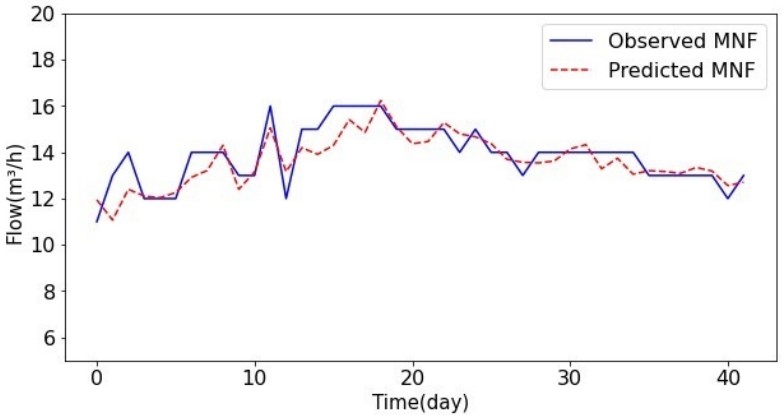

**Figure 12.** MNF prediction with the test data from YD 4 with the LSTM model.

### *3.4. Result Summary with MLP-MNF and LSTM-MNF*

The algorithm used for this result was applied to the YD 2, YD 3, and YD 4 DMAs. Mean Absolute Error (MAE), Mean Square Error (MSE), Mean Absolute Percentage Error (MAPE), and R-Squared were used as evaluation criteria. As shown in Table 3, the average absolute ratio errors for each DMA showed good predictive power as a result of predicting through YD 2 (6.231%), YD 3 (3.682%), and YD 4 (3.806%). The R-squared is the degree of contribution of the parameters with significant difference in each subgroup. $R^2 \geq 0.7$, $0.7 > R^2 \geq 0.5$, $0.5 > R^2 \geq 0.3$, and $0.3 > R^2$ were considered high, moderate, low, and very low values, respectively [14]. The reason for the relatively low R-squared result of YD 3 might be that it was highly variable and noisy compared to YD 2 and YD 4.

**Table 3.** Evaluation results for three sectorized DMAs (YD 2, YD 3, YD 4).

| Algorithm | YD 2 Data | | | |
|---|---|---|---|---|
| | **MAPE** | **MAE** | **MSE** | **R-Squared** |
| MLP-MNF | 7.572 | 0.372 | 0.612 | 0.868 |
| LSTM-MNF | 6.231 | 0.416 | 0.259 | 0.907 |
| **Algorithm** | **YD 3 Data** | | | |
| | **MAPE** | **MAE** | **MSE** | **R-Squared** |
| MLP-MNF | 4.020 | 0.849 | 1.135 | 0.257 |
| LSTM-MNF | 3.682 | 0.751 | 1.010 | 0.488 |
| **Algorithm** | **YD 4 Data** | | | |
| | **MAPE** | **MAE** | **MSE** | **R-Squared** |
| MLP-MNF | 4.253 | 0.581 | 0.548 | 0.637 |
| LSTM-MNF | 3.806 | 0.532 | 0.419 | 0.733 |

## 4. Discussion

The model proposed in this study can be used as an early warning system that will strengthen waterwork network management. When the prediction results show that the MNF increases quickly, we can take preemptive actions promptly—such as repair and replacement—so as to minimize losses, e.g., from water leakages from external forces such as road works.

It can be seen that the accuracy of the LSTM-MNF model's predictions with the YD 2, 3, and 4 data was better than the MLP-MNF model. The MAEs of the LSTM-MNF model for YD 2, 3, and 4 were 0.416, 0.751, and 0.532, respectively. The MSEs of the LSTM-MNF model for YD 2, 3, and 4 were 0.259, 1.01, and 0.419, respectively. The MAPEs of the LSTM-MNF model for YD 2, 3, and 4 were 6.231, 3.682, and 3.806, respectively.



There are many improved versions of these artificial intelligence models, such as Bi-directional Long Short-Term Memory networks (Bi-LSTMs) and Gated Recurrent Units (GRUs). These and other new methods can be used for comparison with the model proposed for improved prediction.

## 5. Conclusions

This study proposed the LSTM-MNF prediction model based on a neural network using the water flow data of three DMAs in County Y. The flow data were collected from the SCADA system via TM/TC. Then, machine learning approaches such as LSTM and MLP were implemented to predict the water consumption of the next day with the training from the previous 7 days. MNF data were then derived from the predicted flow.

When evaluating the performance of the LSTM-MNF and MLP-MNF models, the results showed that the LSTM-MNF model proposed in this study showed better prediction than the MLP-MNF model. This is because of its characteristic of remembering patterns in time series for long durations of time. LSTM networks are specific types of Recurrent Neural Network architecture that can use long-term data to predict future. The LSTM model can better take into account the time-dependent structure of water data.

The study not only provides prediction flow, but also acts as an important support for decision-making in terms of repair and replacement, identifying the time at which water leakages start to occur. It is judged that it will be of great help in detecting and reducing the amount of water leakage through the prediction of the minimum flow at night. It is also possible to replace the missing flow data with the prediction flow data in a timely manner.

Due to the difficulty of acquiring data for more than three places, we could only apply the model for MNF prediction to the YD 2, 3, and 4 DMAs, which cannot reflect the entire network of County Y, which includes 21 DMAs. Additionally, the method proposed in this study may not be effective for customers who have big water tanks, such as fire stations, hospitals, night markets or apartments complexes, which may use a lot of water at night. This pattern of water demand can make it difficult to estimate the MNF and the exact amount of leakage if a lot of water is used at night for business activities. We suggest that this problem can be solved by installing many flow meters in sub-DMAs. Flow meters are mainly installed in a DMA to measure the water of 500~1500 households. In order to establish more accurate predictions, we suggest that many flow meters be installed to monitor the water flow, e.g., installing flow meters for every 100 households. In addition, the proposed model could not predict MNF after one week; the performance of the model will be improved by training data for long periods in the future. More accurate MNF can be predicted in the future by considering additional data, such as weather information.

**Author Contributions:** S.S.L. provided the problem of prediction design with LSTM and the machine learning algorithm. H.-H.L. proposed the data analysis. Y.-J.L. advised the method of the proposed algorithm and contributed to sentence correction. All authors provided substantive comments. All authors have read and agreed to the published version of the manuscript.

**Funding:** Kyungpook National University Research Grant.

**Institutional Review Board Statement:** Not applicable.

**Informed Consent Statement:** Not applicable.

**Data Availability Statement:** Not applicable.

**Conflicts of Interest:** The authors declare no conflict of interest.

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
