# Peer review of "Prediction of Minimum Night Flow for Enhancing Leakage Detection Capabilities in Water Distribution Networks"

_applsci, doi:10.3390/app12136467_

Round 1

Reviewer 1 Report

  • A review of related work must be included.
  • Other statistics that account for the dynamics prediction capability of the models should be included, for example Nash index, Variance Accounted For (VAF) and Percentage Of Change In Direction (POCID) among others.

Author Response

Upload the comments.

Reviewer 2 Report

According to the rports of the authors, the answering was interesting and completely.

Author Response

Thank you for the review.

Reviewer 3 Report

Manuscript ID: applsci-1729252

Title: Prediction of minimum night flow for the leakage detection in water distribution networks

Authors: Sang Soo Lee, Ho-Hyun Lee and Yun-Jung Lee

This paper presents a case study where a model-based approach with a Multi-Layer Perceptron (MLP) and Long Short-Term Memory (LSTM) was used to forecast the minimum night flow for selected DMA in a county in South Korea. Authors applied (as-is) readily available solutions in the KERAS Python library. 

It is sound applied research of minimum scientific impact but concrete, practical benefits for interested readers, based on real (and not perfect) data.

Specific remarks:
1. In general, Authors should reread the text for language clarifications and rewrite it for clarity (cf. page 1, lines 32-33, the caption for Fig. 1, 2 and 3, page 2, lines 44-45 and 54-56, page 3 line 78, page 7 lines 147-151).
2. The introduction is concise, but the Authors should at least expand the section on NRW statistics. Currently, only South Korea is used as an example - how does it compare to other countries? (e.g. in the top 5 and bottom 5 in terms of NRW). What methods are used to minimise NRW in South Korea?
3. Page 1, lines 32-33 - I do not understand this sentence - please rewrite or delete it.
4. Figure 1 is not cited in the body of the paper. Please comment or delete. Please correct the typo in the Figure caption.
5. Page 2, lines 38-40: "The water demand at night includes household toilet usage, factories, hospitals, etc. The remaining amount is judged as the leakage amount." This statement is a vast simplification; the Authors should more clearly describe the methodology of extracting leakage flow for MNF analyses.
6. Page 2, lines 50-51: Please explain the abbreviations first.
7. Page 2, lines 54-55: Please provide more quantitative information about target DMAs (e.g. size in area and number of customers, profile/composition of water consumers, min / max / average water consumption etc.).
8. Page 2, lines 54-55: What kind of household water metering is in place for target DMAs (smart metering? If yes - what is the resolution of data obtained?)
9. Page 2, lines 64-65: Authors should define a "relatively" high-accuracy electronic (electromagnetic?) flow meter. Please provide the brand and model of used measuring equipment (this will allow readers to assess source data quality properly).
10. Page 2, lines 64-65: Please comment on how this flow meter accurately measures low flows at night and large flows during peak demand hours in DMA?
11. Page 3, lines 70-71: Please comment on why the Authors only used only half a year's worth of data?
12. Page 3, lines 70-71: Why did the Authors skip 16-31 December?
13. Page 3, lines 70-71: Did the observation period include any national holidays?
14. Page 3, line 71: Please comment how representative it is to replace (for example) data from 5 am Monday (working day) with 5 am Sunday (free day).
15. Page 3, line 71: Please comment on how the Authors checked data for consistency and other methods used to wrangle and preprocess time series data.
16. Page 3, line 79: Please comment on why the Authors selected a 1-hour interval? Mains powered flow meters provide data in seconds (a short comment on page 7 is not enough, as it does not address the reason for using a large data interval).
17. Page 3, line 81: Please explain the used abbreviations first.
18. Page 4 - Figures - Please correct flow rate units in the figures to standard m3/h.
19. Page 4 - Figures - Why did the Authors select an observation period of 40 days to show data? Can you provide actual dates (a reader would like to know the month/day) 
20. Page 4 - Figures - Are the observation periods related to the same dates?
21. Page 6 table 2: Please comment on relatively low R-squared result for YD 3 and MLP (R2 = 0.257) and LSTM (R2 = 0.488) method. What was the reason for such large deviation comparing to YD 2 (R2 > 0.8) and YD 4 (R2 > 0.6)?
22. Page 7, line 153: Explain GRU abbreviation.
23. Page 7, line 170: Authors should explain more what it means: […] that many flow meters be installed in various places to monitor the water flow?
24. Page 6-7 Discussion: Authors should provide a more in-depth explanation of why LSTM was performing better than MLP. The discussion is minimal in its current form and requires significant improvements.
25. Page 6-7 Discussion line 147-150. Authors should move suggestions for improvements at the end of the conclusions, where Authors can clarify the planned next steps of this research.

Author Response

Thank you for the review.
